# LTLBench: Towards Benchmarks for Evaluating Temporal Logic Reasoning in Large Language Models

## Abstract

Temporal reasoning (TR) is a critical component of artificial intelligence, encompassing understanding and processing temporal information and relationships between events. To discover and study the TR ability in Large Language Models (LLMs), various datasets have been constructed in different ways for evaluating various aspects of TR ability. Our work proposes a novel approach to design and develop a pipeline for constructing datasets to evaluate the TR ability of LLMs by leveraging random directed graph generation, LTL formula, and the NuSMV model checker. Based on the pipeline, we have also constructed a dataset as a benchmark, namely LTLBench, consisting of 2,000 TR challenges and evaluated six LLMs with it. Furthermore, we have conducted additional experiments to discover the impact of increasing the number of events and formula operators on the complexity of TR problems and the performance of LLMs. We have demonstrated that although LLMs exhibit some promise in handling TR challenges, they still struggle with complex TR. We expect this work can offer insights into TR ability in LLMs while also providing a valuable tool for future TR evaluations.[1]

## 1 Introduction

Temporal reasoning (TR) is a fundamental and critical aspect of artificial intelligence, that encompasses understanding, processing, and reasoning about the temporal information and relationships between events, which is essential for handling and solving problems in various scenarios (Shoham & Goyal, 1988; Chittaro & Montanari, 2000; Vila, 1994). Recently, Large Language Models (LLMs) have demonstrated and shown promise and emergence of various reasoning abilities, including but not limited to mathematical reasoning (Gaur & Saunshi, 2023; Wei et al., 2022; Imani et al., 2023), commonsense reasoning (Bian et al., 2024; Wei et al., 2022), and theory of mind reasoning (Strachan et al., 2024; Kosinski, 2023; Tang & Belle, 2024a;b). Nevertheless, there still remains a lack of definitive consensus regarding the emergence, performance, and robustness of their ability to tackle TR challenges. Several studies have demonstrated that although LLMs have shown some promise with TR ability, they still struggle with TR and there is a substantial gap on the performance of handling and solving TR challenges between the state-of-the-art LLMs and humans (Chu et al., 2023; Wang & Zhao, 2024; Beniwal). Furthermore, since TR ability is crucial for handling and processing temporal information and relationships between temporal events and TR problems are ubiquitous among many activities (Vila, 1994), for LLMs, in their frequently mentioned and utilized scenario, question and answering, correctly comprehending temporal information and handling temporal tasks is necessary and warranted to provide efficient and helpful responses. In addition, LLMs are also discussed and used in various ways such as embodied agent (Liu et al., 2023b; Wang et al., 2023; Liu et al., 2023a) for which planning and decision making unavoidably intertwine with TR. Therefore, it is necessary and important to study and discover the TR ability of LLMs.

In order to discover and evaluate the TR ability of LLMs, various benchmarks have been developed in previous studies using various dataset construction approaches, aiming at evaluating the TR ability of LLMs from different perspectives and at different levels of complexity. For example, in Xiong et al. (2024), they utilize

---

[1]The source code of the study is open source and available at `https://anonymous.4open.science/r/LTLBench-DF3E`.

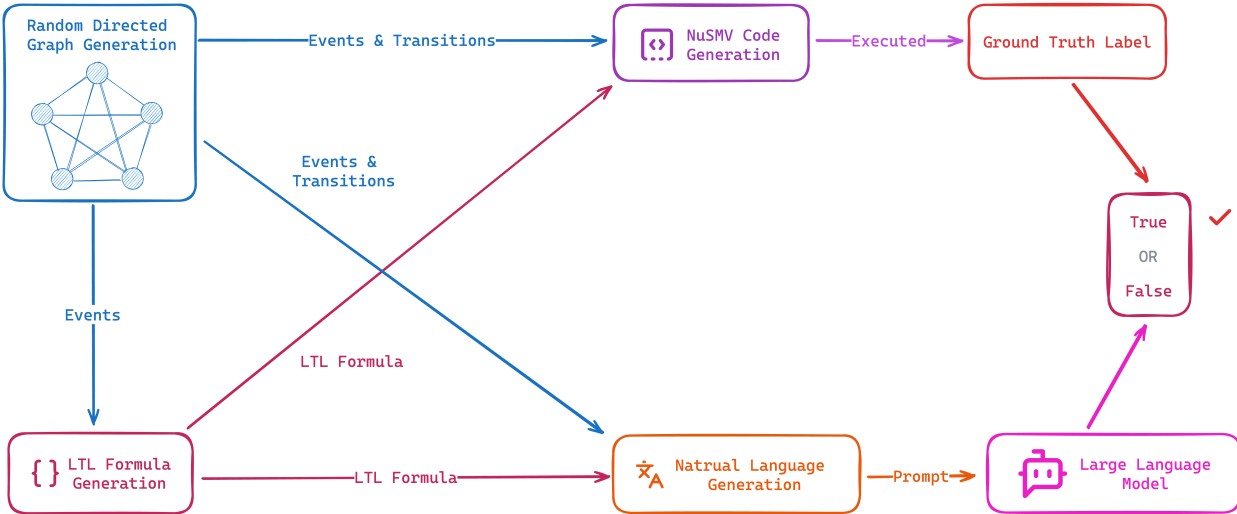

Figure 1: The overview of the TR problem generation pipeline.

GPT-3.5 to generate graph-based TR problems based on the temporal knowledge graph YAGO11k (Dasgupta et al., 2018). Additionally, in Fatemi et al., they employ random graph generation as a foundation and preparation to form rule-based and different types of temporal facts and questions. In order to further discover the TR ability of LLMs and push the boundaries of evaluation of TR ability, we proposed and designed a novel pipeline for TR dataset construction which allows for the controllable and scalable generation of TR problems at any level of complexity and size. Based on it, we have also generated and constructed a dataset consisting of 2,000 TR problems as a benchmark, namely LTLBench.

Inspired by Fatemi et al., although our approach also involves random graph generation as the prerequisite and preparation for subsequent problem generations, the methodology differs a lot from it in terms of the usage of generated graphs and subsequent problem generations. In our approach, a generated TR problem mainly consists of a context that depicts the situation of a TR problem and a hypothesis requiring LLMs to judge its validity against the given context of the problem. In addition, the core components in the process of dataset generation involve a randomly generated directed graph which forms the preparation for subsequent problem generations, a random linear temporal logic (LTL) formula that serves as the hypothesis regarding a given problem context, and the NuSMV model checker (Cimatti et al., 2002) which allows for running the code which represents the events, the transitions between events, and also the LTL formula to provide the ground truth label for the TR problem.

It is worth noting that although complex LTL is always discussed and used in the context of formal and program verification, basic and reasonably complex LTL tasks are ubiquitous in many daily tasks. For example, if people are out of milk, they will eventually buy it, which can be formalized in an LTL formula as $G(\text{OutOfMilk} \rightarrow F(\text{BuyMilk}))$, and also if the traffic light is green, it will then turn to yellow, which can be formalized as $G(\text{Green} \rightarrow X(\text{Yellow})$.[2] Therefore, LLMs are supposed to handle reasonably complex LTL problems in order to provide efficient and accurate responses. Therefore, we propose to use LTL, a subset of temporal logic, to form reasonably complex hypotheses of TR problems.

In addition, as shown in Figure 1, during the generation process for a TR problem, we first generate a random directed graph. Then, we adopted and slightly modified the LTL formulas generation algorithm designed by Zhu (2021) to generate an LTL formula based on the events given in the graph. Subsequently, we convert the information of events given in the graph and the LTL formula into NuSMV (Cimatti et al., 2002) code and execute the code to obtain the ground truth label of the TR problem. Finally, the information of events and the LTL formula are converted into the TR problem in the form of natural language.

---

[2]Refer to Kröger & Merz (2008) and Goranko & Rumberg (2024) for more details on LTL syntax and semantics.

Furthermore, in order to intensively and comprehensively evaluate LLMs, we selected six models of which three are the models with large parameter sizes and three are the models with small parameter sizes, and evaluated them on LTLBench consisting of 2,000 generated TR challenges. We demonstrated that although LLMs have shown promise in handling TR challenges, they still struggle with complex TR. In addition, we have also taken additional experiments to discover how the number of events and operators may affect the complexity of TR problems and the performance of LLMs. The key contributions of our study are outlined as follows:

1. We have designed and developed a novel pipeline based on the random directed graph generation, LTL formula, and the NuSMV model checker for TR problem generation and dataset construction. Based on it, we have also generated and constructed a dataset, LTLBench, as a benchmark for evaluating the TR ability of LLMs. The data generation process in the pipeline is controllable and scalable, meaning it enables and allows for generating TR problems at any level of complexity and size;

2. We have also taken intensive and comprehensive experiments on six selected models for which three are models with large parameter sizes and three are in small parameter sizes and demonstrated that they still face significant challenges when tackling complex TR problems.

## 2 Related Work

### 2.1 TR in LLMs

Temporal reasoning has recently obtained substantial attention and study (Xiong et al., 2024; Fatemi et al.; Beniwal; Chu et al., 2023; Hu et al.; Liu et al.; Vashishtha et al., 2020). In the work of Beniwal, they point out the deficiencies of LLMs in terms of their ability to understand and handle temporal information and reasoning. Additionally, in Xiong et al. (2024), they introduce and propose a framework, TG-LLM, to improve the performance of LLMs in tackling TR tasks. Nevertheless, the evaluation and enhancement of TR ability in LLMs are still in progress and need more effort and study.

### 2.2 TR Benchmarks

To evaluate TR ability of LLMs, various TR datasets and benchmarks have been developed and constructed in different approaches for evaluating different aspects of TR ability of LLMs at varying levels of complexity (Fatemi et al.; Wang & Zhao, 2024; Xiong et al., 2024; Beniwal; Qin et al., 2021; Tan et al., 2023; Virgo et al.). For instance, in Xiong et al. (2024), a TR dataset is constructed by leveraging a large temporal knowledge graph, YAGO11k (Dasgupta et al., 2018), and utilizing GPT-3.5 and rule-based Python script to generate TR challenges. However, in the work of Fatemi et al., they leverage graph generation and use rule-based methods without introducing LLMs to generate TR tasks, highlighting and focusing on temporal semantics and arithmetic reasoning. Additionally, in Wang & Zhao (2024), they introduce a TR dataset consisting of various temporal aspects such as order, arithmetic, frequency, and duration. Our work introduces a novel pipeline for generating TR problems based on graph generation, LTL formula, and the NuSMV model checker, allowing for the control of the complexity and size of problems.

## 3 TR Problem Generation Pipeline

The pipeline to generate a single TR problem consists of four stages: (1) Random Directed Graph Generation, (2) LTL Formula Generation, (3) NuSMV Code Generation, and (4) Natural Language Generation. Figure 1 demonstrates the overview of the process for a TR problem generation.

### 3.1 Random Directed Graph Generation

During this stage, a directed graph is randomly generated with a given number of events $n$ where $n > 1$ to ensure the formation of transitions between events.

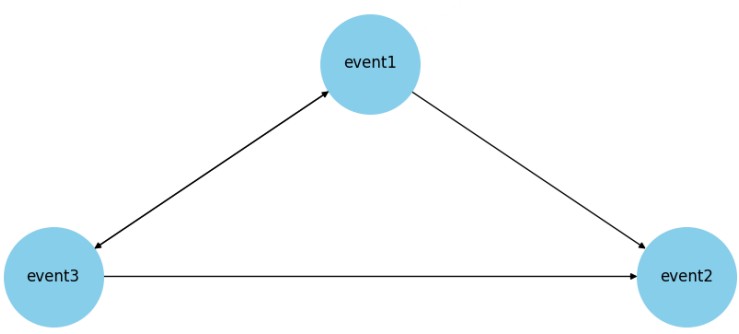

Figure 2: An example of a generated random directed graph.

In this graph, each $node_i$ represents an individual $event_i$, and each $edge_j^i$ is a directed edge pointing from $node_i$ to another $node_j$, which forms the relationships and transitions between events, indicating that $event_j$ represented by $node_j$ can occur after the $event_i$ represented by $node_i$. It is important to note that each $node_i$ within the graph can have multiple outgoing edges, signifying that several subsequent events can follow $event_i$, as well as multiple incoming edges, indicating that $event_i$ can be preceded by several other events.

As an example illustrated in Figure 2, given $n = 3$, three events are generated: $event_1$, $event_2$, and $event_3$. The case that $event_1$ points to $event_2$ indicates that $event_2$ can happen after $event_1$. The case that $event_1$ points to $event_3$ and $event_3$ also points to $event_1$ means that $event_1$ can happen after $event_3$ and $event_3$ can happen after $event_1$. In addition, $event_1$ not only points to $event_2$ but also $event_3$, indicating that both $event_2$ and $event_3$ can happen after $event_1$.

The generation of the random directed graph during this stage serves as the foundation and preparation for generating the LTL formula, NuSMV code, and TR problem represented in natural language by providing the information of events and also the transitions between events.

### 3.2 LTL Formula Generation

Based on the events generated in the graph, we employed and slightly modified the algorithm designed by Zhu (2021) to generate an LTL formula[3] with a given number of operators $m$ where $m > 0$. The LTL operators include unary and binary operators. Unary operators, for example, include but are not limited to $X$ which indicates that for a given event $\phi$, $X\phi$ denotes that the event $\phi$ will occur at the next moment, and $F$ for which $F\phi$ means that event $\phi$ will eventually occur at some point in the future. Binary operators include but are not limited to & representing logical AND and | representing logical OR. The given number of operators refers to the number of unary and binary LTL operators contained in an LTL formula. The algorithm for generating LTL formulas is detailed in Algorithm 1.

An example of a generated LTL formula is also provided and shown in Listing 1. The formula means that if $event_1$ happens, it will be globally true that, at some point in the future, $event_2$ will eventually happen.

```
(event1 -> (G (F event2)))
```

Listing 1: An example of a generated LTL formula.

The generation of the LTL formula aims to be the preparation for generating the LTLSPEC part of the NuSMV code and also to generate the hypothesis part of the TR problem represented in natural language.

---

[3]Note: only one LTL formula is generated for each given graph from the previous stage, meaning the $formulaCount$ parameter in Algorithm 1 is always 1.

---

**Algorithm 1** Generate LTL Formulas

---

1: **procedure** GENERATELTLFORMULAS($states, formulaLength, formulaCount$)
    **Input:** $states$ - an array of atomic states representing events
    **Input:** $formulaLength$ - the number of operators
    **Input:** $formulaCount$ - the number of formulas to generate
    **Output:** an array of LTL formulas
2:     $unaryOperators \leftarrow [X, G, F, !]$
3:     $binaryOperators \leftarrow [\&, |, \rightarrow]$
4:     $operators \leftarrow unaryOperators + binaryOperators$
5:     $B \leftarrow [[ \ ] \text{ of size } formulaLength + 1 \ ]$
6:     $B[0] \leftarrow states$
7:     $formulas \leftarrow [ \ ]$
8:     **for** $i \leftarrow 1$ **to** $formulaCount$ **do**
9:        **for** $j \leftarrow 1$ **to** $formulaLength$ **do**
10:          $x \leftarrow$ randomly choose an operator from $operators$
11:          **if** $x \in unaryOperators$ **then**
12:            $y \leftarrow$ randomly sample a formula from $B[j-1]$
13:            $newFormula \leftarrow [x, y]$
14:          **else**
15:            $s \leftarrow$ randomly sample a integer from $[0, j)$
16:            $y1 \leftarrow$ randomly sample a formula from $B[s]$
17:            $y2 \leftarrow$ randomly sample a formula from $B[j-1-s]$
18:            $newFormula \leftarrow [y1, x, y2]$
19:          **end if**
20:          append $newFormula$ to $B[j]$
21:        **end for**
22:        $formula \leftarrow B[formulaLength][-1]$
23:        append $formula$ to $formulas$
24:     **end for**
25:     **return** $formulas$
26: **end procedure**

---

### 3.3 NuSMV Code Generation

Given the information of events from the graph and the LTL formula, this stage aims at converting and representing them in NuSMV code. The generation of the NuSMV code is divided into two parts: (1) context generation and (2) LTLSPEC generation. The context describes the situation of the TR problem while LTLSPEC represents a hypothesis regarding the context.

For the context generation, it includes event definitions, initial event setup, and event transitions setup. Based on the generated graph, the events and their transitions are converted into the context part of the NuSMV code while an initial event is set up randomly and also converted into the NuSMV code. As shown in Listing 2, Lines 2-3 define the three events, Line 5 sets up the initial event which is $event_3$, and Lines 6-12 construct the transitions of events in which, for instance, Line 7 shows that $event_2$ can follow $event_1$. In addition, Line 11 indicates that $event_2$ remains to itself and after $event_2$, no other events can happen, if there is no other transition specified from $event_2$ to $event_i$ where $i \neq 2$.

For LTLSPEC generation, we convert the generated LTL formula into code that complies with NuSMV, as illustrated at Line 13 in Listing 2 which represents the LTL formula shown in Listing 1.

The generation of the NuSMV code aims at providing the ground truth label for the TR problem. The generated code will be executed by the NuSMV model checker to give the ground truth label during the generation process.

```
1  MODULE main
2  VAR
3      state : {event1, event2, event3};
4  ASSIGN
5      init(state) := event3;
6      next(state) := case
7          state = event1 : event2;
8          state = event1 : event3;
9          state = event3 : event1;
10         state = event3 : event2;
11         state = event2 : event2;
12     esac;
13 LTLSPEC ((state=event1) -> (G (F (state=event2))))
```

Listing 2: An example of NuSMV code.

```
1  === Context ===
2
3  Initially, event3 happened. After event1, event2 can happen. After event1, event3 can happen
   ↪ . After event2, no other events can happen. After event3, event1 can happen. After
   ↪ event3, event2 can happen.
4
5  === Hypothesis ===
6
7  C1: Event2 will happen eventually.
8  C2: C1 will always be true at any future time.
9  C3: That event1 happens implies that C2 holds.
10
11 C3 is True or False? Answer with "True" or "False" directly:
```

Listing 3: An example of TR problem in the form of natural language.

### 3.4 Natural Language Generation

During this stage, the events information in the graph and the LTL formula will be converted and represented in natural language. Similar to the NuSMV code generation, this stage can also be divided into two parts: (1) context generation and (2) hypothesis generation. The context describes the situation of the problem while the hypothesis is what the LLMs need to judge for validity against the given situation.

For context generation, based on the generated graph and the initial event, we convert the information of events and their transitions into natural language. As shown at Lines 1-4 in Listing 3, the initial event and events transitions are converted and described in natural language.

For hypothesis generation, we transform the generated LTL formula into natural language as shown at Lines 5-10 in Listing 3. Additionally, Line 11 is used to prompt LLMs to judge the validity of the hypothesis, requiring an answer in "True" or "False" format.

The TR problem represented in natural language generated in this stage is the core and final product of the generation process, which will be used to evaluate the TR ability of LLMs.

## 4 Experiment Settings

To evaluate the TR ability of LLMs, based on the pipeline, we generated and constructed a dataset, LTL-Bench, consisting of 2,000 problems. Each problem features the number of events $n = 3$ and the number of formula operators $m = 3$. Additionally, to explore the impact of changes in the number of formula operators, we conducted evaluations on newly generated problems with the fixed number of events $n = 2$ while varying the number of formula operators for which $m \in \{1, 2, 3, 4, 5, 7, 9\}$. For each $(n, m_i)$, such as $(2, 1)$ indicating that the number of events is 2 and the number of operators is 1, there are 300 problems generated as a dataset for evaluation. Similarly, we fixed the number of formula operators $m = 2$ and varied the number of

Table 1: The metrics of LLMs on LTLBench

| Models | Accuracy | F1 | AUC |
|---|---|---|---|
| *gpt-3.5-turbo* | 0.56 | 0.55 | 0.56 |
| *llama3:70b-instruct* | 0.59 | **0.59** | 0.59 |
| *qwen:72b-chat* | **0.60** | **0.59** | **0.60** |
| *gemma:7b-instruct* | 0.55 | 0.53 | 0.55 |
| *qwen:7b-chat* | 0.54 | 0.54 | 0.54 |
| *mistral:7b-instruct* | 0.54 | 0.50 | 0.54 |

events for which $n \in \{2, 3, 4, 5, 7, 9\}$ to examine the effects of changing the number of events while keeping the number of operators constant. In addition, for all generated datasets, their distributions of ground truth labels are meticulously balanced, meaning half of the problems are labeled as *True* while half of the problems are labeled as *False*.

Furthermore, for comprehensive evaluations, we selected six models. Three of them are in larger parameter sizes including GPT-3.5 Turbo noted as *gpt-3.5-turbo*, Llama 3 for which we choose its instruct model with 70 billion parameters noted as *llama3:70b-instruct*, and Qwen (Bai et al., 2023) for which we utilize its chat model with 72 billion parameters noted as *qwen:72b-chat*. In addition, we also choose three models in smaller parameter sizes comprising Gemma (Team et al., 2024) for which we use its instruct model with 7 billion parameters noted as *gemma:7b-instruct*, Mistral (Jiang et al., 2023) for which we choose its instruct model with 7 billion parameters noted as *mistral:7b-instruct*, and Qwen (Bai et al., 2023) for which we use its chat model with 7 billion parameters noted as *qwen:7b-chat*. This selection is intended for assessing the performance variability of handling TR challenges across models with different parameter sizes. In addition, the temperature hyperparameter for all selected models is set to 0 to keep the responses deterministic and also to ensure that the models consistently select the optimal next token.

## 5 Results and Analyses

### 5.1 Evaluation with LTLBench

We have evaluated the selected models on LTLBench which consists of 2,000 generated TR problems with the number of events $n = 3$ and the number of formula operators $m = 3$. The metrics consisting of Accuracy, F1, and AUC are shown in Table 1.

From the metrics, all Accuracy and AUC of LLMs are above 0.5. In addition, the model, *qwen:72b-chat*, demonstrates the highest Accuracy at 0.60, F1 at 0.59, and AUC at 0.60 while the *llama3:70b-instruct* also shows a similar performance. It may indicate that they show promise and ability to handle TR challenges. Nevertheless, the modest metrics signify that they still struggle with complex TR problems.

In addition, the averages of Accuracy, F1, and AUC for the models with large parameter sizes are 0.58, 0.58, and 0.58 respectively, while for the models with small parameter sizes, they are 0.54, 0.52, and 0.54 respectively. It unsurprisingly indicates that models with large parameter sizes may surpass the models with small parameter sizes when handling these complex TR challenges, although the difference between the performance is modest.

### 5.2 Impact of increasing $m$

The selected models are also evaluated on the additional constructed datasets, where the number of events is fixed to $n = 2$ while the number of formula operator $m$ increases from 1 to 9, specifically $m \in \{1, 2, 3, 4, 5, 7, 9\}$. The performance in terms of Accuracy and AUC of the evaluated models is shown in Figure 3.

As shown in Figure 3a, the accuracy of models significantly drops from $m = 1$ to $m = 2$ regardless of the models with large parameter sizes or models with small parameter sizes. It is also the same case for

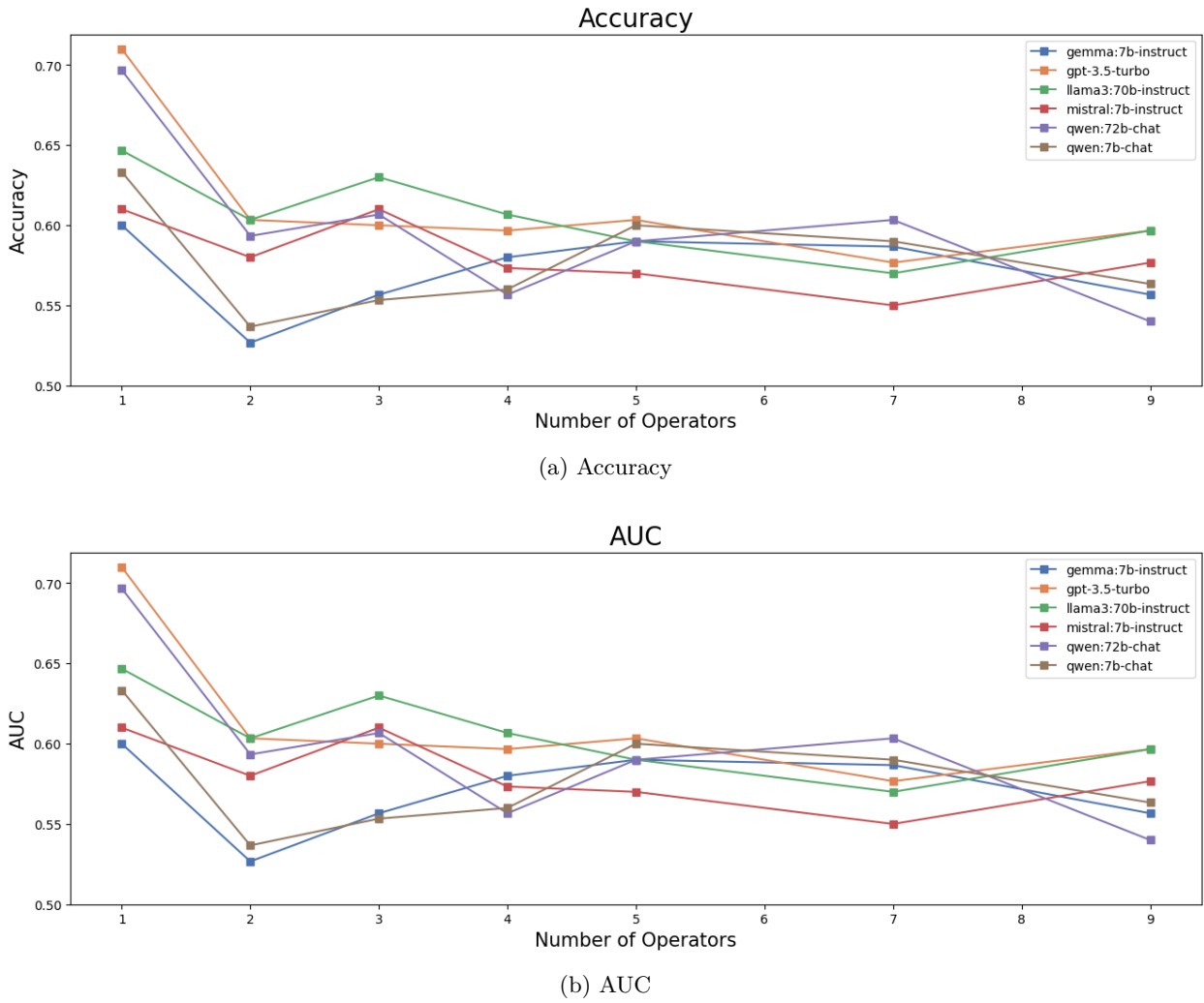

(a) Accuracy

(b) AUC

Figure 3: Accuracy and AUC when increasing the number of operators.

AUC as shown in Figure 3b. Although several models can have a good performance when $m = 1$ such as *gpt-3.5-turbo* and *qwen:72b-chat*, they suffer from TR tasks when $m$ increases. It indicates that increasing the number of formula operators may significantly introduce more complexity of TR and the TR ability of LLMs is currently still lacking consistency and robustness when TR tasks become more complex.

Furthermore, it is obvious to note that models with large parameter sizes outperform all other models with small parameter sizes when $m = 1$. However, they show indifferent performance when $m$ increases while the AUC decreases and approaches to 0.5, signifying that all models start random guessing.

### 5.3 Impact of increasing $n$

In addition, the selected models are also evaluated with the number of formula operators fixed to $m = 2$ while the number of events $n$ where $n > 1$ increases from 2 to 9, specifically $n \in \{2, 3, 4, 5, 7, 9\}$. The performance in terms of Accuracy and AUC of the evaluated models is shown in Figure 4.

As shown in Figure 4a and Figure 4b, the Accuracy and AUC of models show a trend to decrease while the number of events increases. Although it is not as obvious and significant as the case when $n$ is fixed and $m$ increases, it still indicates that increasing the number of events will also introduce a certain level of complexity. In addition, as the same case discussed in Section 5.2, as the number of events increases, models

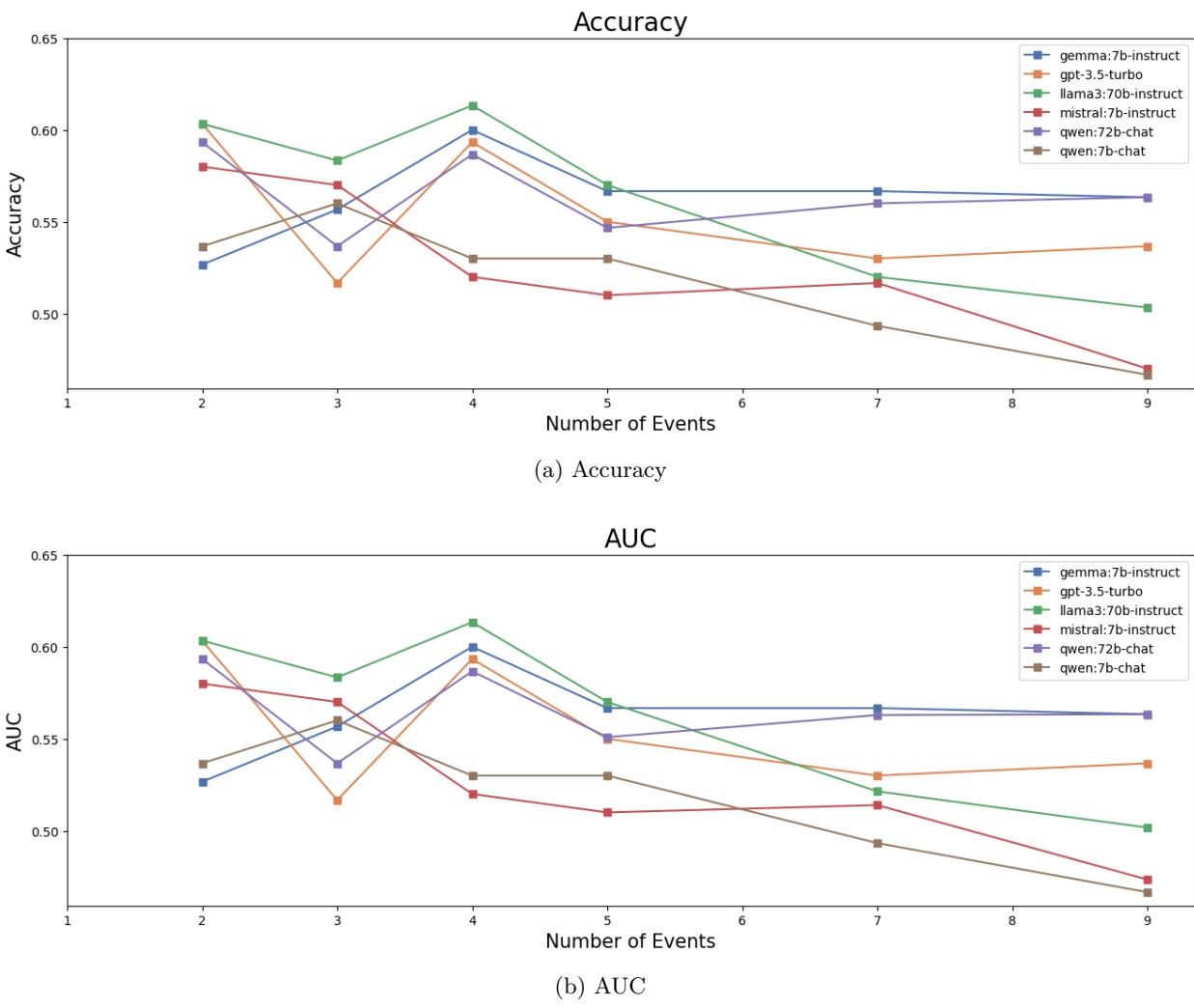

(a) Accuracy

(b) AUC

Figure 4: Accuracy and AUC when increasing the number of events.

with large parameter sizes show indifferent performance to the models with small parameter sizes since their AUC approaches 0.5, indicating random guessing.

# 6 Implications

We designed a pipeline for constructing TR datasets to evaluate the TR ability of LLMs. The pipeline is controllable and scalable to be able to introduce any level of complexity of TR by modifying the number of events and the number of formula operators. Future work can leverage this pipeline to generate TR problems at a certain level of complexity depending on specific needs.

In addition, since TR is crucial in many scenarios, we hope this work can not only offer insights into the TR ability of LLMs but also provide a valuable tool to further evaluate the TR ability of future LLMs or other AI systems that are required to equip with a certain level of TR ability, thus, to enable confident deployment and application.

# 7 Limitations and Future Work

Considering the difficulty of generated TR problems, we only included a subset of LTL operators in the TR problem generation pipeline but excluded several LTL operators such as $U$ (Until) and $R$ (Release). However, adding and incorporating more LTL operators should be easy and straightforward to enable future work to evaluate the more complex TR ability of LLMs.

In addition, although we have intensively and comprehensively evaluated six models, due to the unavoidable heavy computation costs, we did not run the evaluation on GPT-4 and other LLMs exhaustively. Nevertheless, future work can easily leverage the pipeline and the constructed dateset, LTLBench, in this work to evaluate GPT-4 and other LLMs.

# 8 Conclusion

In this study, we designed a pipeline for generating and constructing the TR datasets based on the random graph generation, LTL formula, and the NuSMV model checker. Based on the pipeline, we have generated a dataset, LTLBench, consisting of 2,000 TR challenges and taken intensive and comprehensive evaluations on it with the six selected models in which three are the models with large parameter sizes while the other three models are in small parameter sizes. We have demonstrated that although LLMs show promise and emergence in handling the TR problem, they still struggle with it. We expect our work to offer insights into the TR ability of LLMs and to provide a valuable tool for TR evaluation while hoping it can pave the way for future more intelligent LLMs or AI systems requiring handling complex TR tasks.

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
