# OpenReview forum: "LTLBench: Towards Benchmarks for Evaluating Temporal Logic Reasoning in Large Language Models"
_TMLR — Rejected by TMLR_

### Review · Reviewer_dQYi · 2025-08-19

**Summary Of Contributions:**

This paper proposes a framework, called LTLBench, to generate synthetic datasets for evaluating
the temporal reasoning capabilities of LLMs. In particular, the framework uses a combination of
(a) randomly generating a directed, static graph $G = (V,E)$, where each node in set $V$ is
interpreted as an event in a temporal process and the directed edges in $E$ denote temporal
dependencies, then (b) an LTL formula $\varphi$ with proposition set $V$ is generated such
that it respects the dependencies implied by $E$, and finally (c) the temporal process implied
by $G$ and $\varphi$ is translated to natural language, which can then be used to evaluate LLM.
Additionally, the study compares the temporal reasoning capabilities of six LLM using data
produced by LTLBench, finding that all investigated LLM struggle with temporal reasoning when it
becomes fairly complex.

**Audience:**

Yes

**Audience Explanation:**

The goal of understanding the temporal reasoning capabilities is intriguing and clearly motivated by the omnipresence of these models now and in the future.

**Claims And Evidence:**

Yes

**Claims Explanation:**

The proposed framework is clearly described and for the crucial part a pseudocode algorithm is provided. Likewise, choosing LTL model checking as a surrogate problem to understand the temporal reasoning capabilities of LLMs is reasonable.

**Requested Changes:**

I'll list my key concerns below, followed by some
detailed comments afterwards:
- The three key steps of LTLBench, especially the first and third, are not defined
sufficiently. Step (a), generating the graph $G$, is only described as "randomly
generating". How exactly does it work? How is the graph sampled? These are questions
that arise and need to be discussed. I guess it would be reasonable to allow for parametrised sampling, depending of the intended temporal reasoning task.
- Likewise, step (c), generating the natural language representation, is not discussed
  properly. Are there any scientifically justified reasons for the translation shown in
  Figure 1? Do the results change with a different representation? I guess it is chosen
  to be similar to the NuSMV syntax, but this does not mean it is a good one for currently
  existing LLM.
- Overall, I think a shortcoming of LTLBench is that it does not allow for any kind of
  control over the graphs or formula in its vanilla form, like only generating LTL
  formula of certain depth or structure. While extensions may be done quickly, such
  things should be discussed or even be part of the framework. In this form, it looks
  rather straightforward.
- Section 2.2 introduces several related works concerned with TR benchmarks.
  I wonder what their shortcomings are and where LTLBench does better. For Fatemi et al.
  it is stated that "... they leverage graph generation and use rule-based methods without
  introducing LLM to generate TR tasks." However, this does not seem to be a shortcoming
  of the data they generated, only that they used it for different models.
- The experiments are conducted on very small graphs and formula. I wonder if there is
a complexity reason for this. This is not discussed, I think, and could potentially be
a limitation of LTLBench.

Such discussions should be added.

Detailed comments:
- p. 1: The citation "Beniwal" appears to be broken.
- p. 2: The citation "Fatemi et al." is styled differently (for example, it lacks the year).
- Listings 2 and 3: The term "Listing" seems unusual.

---

> ### Author Response · Authors · 2025-09-01
>
> Thank you for your time and suggestions!
>
> Below, we provide detailed responses to each of the concerns:
>
> **To Comment 1** (*“The three key steps of LTLBench, especially the first and third…”*):
>
> Thank you for pointing this out. To generate a random graph, in particular, let $n$ be the number of nodes that should be generated. We first have $|V| = n$ nodes generated, thus $V = \\{ v_1, v_2, \dots, v_n\\}$. Then, we generate edges $E \subseteq V \times V$. For all $1 \leq i < j \leq n$, we independently generate two random variables, $X_{ij} \sim Bernouli(p)$ and $Y_{ij} \sim Bernouli(p)$ where $p = 0.5$. Then, for two nodes $v_i$ and $v_j$, if $X_{ij} = 1$, we generate a directed edge $(v_i,v_j) \in E$ and if $Y_{ij} = 1$, we generate a directed edge $(v_j,v_i) \in E$. We will add details of how it works in the revised version.
>
> **To Comment 2** (*"Likewise, step (c), generating the natural language representation, is not discussed properly..."*):
>
> For the concern about providing more details of how NL representation generation works, we agree with this point, and we will add its pseudo code in the revised version for more details.
>
> In addition, the main reason for leveraging this specific form of NL representation discussed in the work is to better align with the semantics expressed by the graph and the LTL formula. For "a different representation", we can discuss them from two perspectives.
>
> First, one could consider alternative NL representations that also align with the semantics of the graph and the LTL formula. However, we currently believe that the adopted representation best captures these semantics. Moreover, among the many possible NL representations, selecting one that is most suitable for LLMs may be challenging. Conversely, we think that LLMs should possess the ability to understand and reason over NL that semantically correctly encodes the intended meaning, or alternatively, use prompt optimization techniques to transform an NL representation into one that is more beneficial for their latent space and distribution, thereby improving performance. Thus, choosing or optimizing NL representation among many alternatives may not be the central focus of this work.
>
> Second, one could adopt multiple different natural languages, e.g., French, Korean, and Chinese, in addition to English, to examine whether LLMs demonstrate consistent temporal reasoning abilities across varied linguistic representations. However, this is not the direction emphasized in the current work while it is attractive and interesting.
>
> **To Comment 3** (*"Overall, I think a shortcoming of LTLBench is that it does not allow for any kind of control over the graphs..."*):
>
> We will detail its granularity and flexibility in the revised version.
>
> **To Comment 4** (*"Section 2.2 introduces several related works concerned with TR benchmarks..."*):
>
> Thanks for pointing this out, we will then detail and emphasize how our approach is different from others and why we need another benchmark, LTLBench, to evaluate LLMs' temporal reasoning abilities.
>
> **To Comment 5** (*"The experiments are conducted on very small graphs and formula..."*):
>
> As discussed in the paper, this pipeline enables to generate almost unlimited size of graphs and LTL formulas, which turn out to be complex temporal reasoning challenges. However, as mentioned in Section 5, current LLMs are still limited to this level of complexity, and future research can easily increase the complexity when the temporal reasoning abilities of LLMs become better.

---

### Review · Reviewer_hrMb · 2025-08-19

**Summary Of Contributions:**

The paper introduces LTLBench, a new benchmark designed to evaluate LLM's ability to reason over Linear Temporal Logic. The authors argue that current evaluation resources are insufficient for assessing temporal reasoning in machine learning models, and they propose a systematic dataset construction and evaluation protocol.

Strengths:
- The topic is relevant.
- The paper is well-written and accessible.
- The resource is made publicly available.


Weaknesses:
- The scope of LTL tasks covered appears limited, raising questions about representativeness of real-world reasoning scenarios.
- Lack of ablation studies or deeper error analysis makes it harder to assess the benchmark’s true diagnostic value.
- There is no motivation/analysis on the choice of the random graph generation, and how it affects the generated benchmak.

**Audience:**

Yes

**Audience Explanation:**

A benchmark explicitly designed for temporal logic reasoning lies at the intersection of machine learning, formal methods, and trustworthy AI. Despite its limitations, the attempt to formalize and standardize evaluation in LTL is timely and relevant, with the potential to stimulate valuable follow-up research.

**Claims And Evidence:**

No

**Claims Explanation:**

The claim that the benchmark is comprehensive for assessing LTL reasoning in neural models is not fully substantiated. While the dataset and evaluation are described, the range of tasks is limited and the evaluation focuses on a narrow set of baseline models. Moreover, the evidence provided is descriptive rather than analytical; for example, the authors show performance numbers but do not investigate why models fail or what types of reasoning are especially challenging. This reduces the clarity and persuasiveness of the claims.

**Requested Changes:**

- Broaden and clarify benchmark coverage. Provide more detail on the generation process and justify the choice of LTL fragments/tasks included. Explain representativeness and limitations explicitly. Motivate the choice of the underlying random graph generator.
- Strengthen experimental evaluation. Test on a wider range of models, possibly including neurosymbolic or logic-augmented architectures. Add ablation studies to show diagnostic capability.
- Error analysis. Include a qualitative analysis of typical model failures, to demonstrate the benchmark’s usefulness in identifying weaknesses.
- Provide clearer documentation of dataset size, diversity, and difficulty levels.
- Discuss potential extensions to other temporal logics (CTL, MTL) to situate the work in a broader landscape.
- Improve clarity in figures/tables to highlight key findings.

---

> ### Author Response · Authors · 2025-09-01
>
> Thanks for your time and review!
>
> We detail the response to each of the concerns as follows:
>
> **To Comment 1** (*"Broaden and clarify benchmark coverage. Provide more detail on the generation process...."*):
>
> Thanks for the suggestion. As discussed with Reviewer dQYi, we will add more details about the generation process, including how the graph is randomly generated and how the formal representation is converted to natural language representation, with precise formulas and pseudo code.
>
> For the representativeness and limitations of this pipeline and benchmark, currently, there is a whole range of papers that are exploring various reasoning situations with LLMs, including first-order logic reasoning, non-monotonic reasoning, dynamic epistemic logic reasoning, math reasoning, common sense reasoning, etc. Although some of them, such as dynamic epistemic logic reasoning, may not seem directly to represent real-world applications/tasks, they can actually be used to model and reason in the theory of mind, and also can be used in robotics and multi-agent systems. It turns out to be a similar situation for this work. Although LTL is mentioned more frequently in formal verification, it can also be used in task planning, multi-agent systems, etc. In addition, due to the synthetic nature of this pipeline with a model checker to ensure the correctness, this approach gives a novel way to easily synthesize an unlimited size of challenges and can also control the level of complexity and generate challenges to an unlimited level of complexity, for evaluating LLMs.
>
> For the motivation of leveraging random graph generation, as discussed in Section 2, graph generation is widely applied to temporal reasoning. In addition, in the logic community, it is a standard way to leverage graph generation to generate random tests and different kinds of formulas.
>
> **To Comment 2** (*"Strengthen experimental evaluation. Test on a wider range of models..."*):
>
> For the range of models, in the work, we have chosen not only models with large parameter size but also models with small size to form a comprehensive understanding of how various LLMs can perform on this challenge task. In addition, up to the time of submission, although there exist some approaches that leverage logic-augmented methods to improve LLMs’ reasoning performance in specific domains (e.g., Logic-LM, which focuses on Logic Programming, First-Order Logic, and Constraint Optimization), we have not found any relatively general or at least temporal-reasoning–oriented neurosymbolic or logic-augmented LLM architectures. Therefore, we did not include such models or methods in our evaluation. Nevertheless, evaluating them would be straightforward. If you have specific models in mind that you believe we should evaluate, we would be very happy to incorporate them into our experiments. In addition, for the ablation studies, in Sections 5.2 and 5.3 we conduct ablations on different numbers of events and formula operators to observe how they affect the performance of these LLMs, and we provide the corresponding results and analysis. If any other ablation study is necessary, we are happy to add them in.
>
> **To Comment 3** (*"Error analysis. Include a qualitative analysis..."*):
>
> The original intent of this work is to provide a pipeline and benchmark that can both scale up and control complexity, in order to evaluate the capability boundaries of LLMs in temporal reasoning in a timely manner whenever a new SOTA LLM is released without losing its effectiveness to be easily beaten by SOTA models, as well as to examine which controllable parameters in the pipeline may affect their performance to varying degrees. However, we also agree that adding qualitative analyses is valuable for observing the main types of errors LLMs make during reasoning.
>
> **To Comment 4** (*Provide clearer documentation of dataset size, diversity, and difficulty levels."*):
>
> We have discussed it in the work in Section 4, and we have also open-sourced the code and given detailed comments and instructions about how to run the pipeline in https://anonymous.4open.science/r/LTLBench-DF3E (which is anonymous).
>
> **To Comment 5** (*Discuss potential extensions to other temporal logics (CTL, MTL) to situate the work in a broader landscape."*):
>
> We will discuss it in the revised version.
>
> **To Comment 6** (*Improve clarity in figures/tables to highlight key findings."*):
>
> We will optimize it in the revised version.

---

> ### Comment · Reviewer_hrMb · 2025-10-16
>
> The author has not responded to my revision.
> Therefore, I recommend that the paper be rejected.

---

> ### Author Response · Authors · 2025-10-16
>
> Dear Reviewer,
>
> We did respond on 1st Sept and just notice that the reader is not set to "everyone" since we responded before this option was available. We just modified it and could you have a look?

---

> > ### Comment · Reviewer_hrMb · 2025-10-16
> >
> > Thanks! now I can see it!

---

> > ### Comment · Reviewer_hrMb · 2025-10-20
> >
> > Thank you for the detailed replies. I appreciate the clarifications and the intention to expand sections on data generation, limitations, and potential extensions. However, several points remain only partially addressed. In particular, the representativeness of the benchmark, the motivation behind the random graph design, and the diagnostic value of the evaluation would benefit from deeper empirical and analytical discussion rather than descriptive explanations. The planned additions are welcome, but the current evidence still appears limited to fully substantiate the main claims.

---

> ### Author Response · Authors · 2025-10-20
>
> Thank you for the constructive and helpful feedback.
>
> Regarding *"the representativeness of the benchmark & motivation behind the random graph design"*, Reviewer oweF also has a similar concern at the Comment 3. In addition to the discussion of related benchmarks for TR problems mentioned in the paper, we will further elaborate on the rationale and motivation for leveraging LTL in TR problem generation, as well as clarify the scope it covers. However, for the introduction of the random graph design in the pipeline, we think it is more like a natural pre-step before generating LTL formula and the motivation of using LTL may be the focus that is needed to be clarified clearer.
>
> Concerning *"the diagnostic value of the evaluation"*, honestly, we think that identifying a general explanation for why LLMs struggle with TR problems can be complex and challenging. Generally speaking, this difficulty of LLMs stems from the probabilistic nature of their decoding and sampling processes, especially when dealing with data generated from systems that are able to generate problems with vast solution spaces, making data out-of-distribution.  Nevertheless, if we uncover any notable insights, we will include a deep qualitative analysis in the revised version.
>
> We appreciate your helpful comments, and regardless of the final decision, we will take your suggestions into account to further strengthen the paper.

---

### Review · Reviewer_oweF · 2025-10-01

**Summary Of Contributions:**

In this work, the authors proposed a novel pipeline for Temporal Reasoning problem generation. The pipeline combines random directed graph generation, LTL formula, and the NuSMV model checker. They also constructed a novel benchmark LTLBench, as a benchmark for
evaluating the TR ability of LLMs from the proposed pipeline. As the pipeline is a synthetic data generation process, it can handle arbitrary complex problems.

Key strength:
- the authors proposed a novel TR generation pipeline with ground truth answered also generated by NuSMV model checker.
- the pipeline can handle any complexity of problems

Key weakness:
- the empirical results in the paper is quite out-dated at this point. With the newest LLM families being GPT 5 and Qwen3 for example.
- there are many points that are unclear in the paper, see detailed comments below.

**Audience:**

Yes

**Audience Explanation:**

The temporal reasoning problem is one of the core challenges of LLM reasoning thus a novel benchmark would be of interest to the community. A well-designed benchmark would showcase the limitation of current LLM models thus helping researchers develop corresponding improvements for them.

**Broader Impact Concerns:**

There is little broader impact concerns to this work, the authors addressed implications in Section 6.

**Claims And Evidence:**

Yes

**Claims Explanation:**

The key strengths and weaknesses of the paper is listed above.
I believe that the authors claims in the paper is largely correct on the problem generation pipeline itself. However, the empirical evidence LLMs are largely outdated at this point thus it is hard to tell if the current generation of LLMs would still benefit of the constructed benchmark or the performance would have saturated.

**Requested Changes:**

I request the following changes from the authors.

Changes critical for acceptance:
- the models tested are quite out-dated, more recent LLM models such as Qwen3 family should be tested and see their difference in performance

Clarification and comments:
- the abstract reads vague and boilerplate, lacks detailed insight and lack motivation for why a new TR problem generation pipeline is needed.
- What is the motivation of specifically using the LTL formula for generating TR tasks, it is not clearly explained. Which families of TR problems can be represented and to what extent?
- What does algorithm 1 line 4 mean? "operators ←unaryOperators+ binaryOperators" do you mean you actually add the operators, how is this defined?
- What does algorithm 1 line 5 mean? what is `[[]`
- Section 3.4 is unclear, how are the events in Listing 3 suitable as a natural language task? Real world relations often have causal dependencies. In section 1, the authors even gave the example of OutOfMilk --> BuyMilk causal relation. These might be well-captured by LLMs, however by simply feeding event3, event2, event1 etc. into LLMs, it is not considering any real world semantic meaning.

---

> ### Author Response · Authors · 2025-10-02
>
> Thanks for your review!
>
> We elaborate the response to each of the concerns as follows:
>
> **To Comment 1** (*"The empirical results in the paper is quite out-dated at this point. With the newest LLM families being GPT 5 and Qwen3 for example." & "The empirical evidence LLMs are largely outdated at this point thus it is hard to tell if the current generation of LLMs would still benefit of the constructed benchmark or the performance would have saturated"*)
>
> The central focus of this work is to provide an extensible and complexity-controllable pipeline for generating TR challenges, which facilitates the evaluation of both current and future LLMs on TR reasoning. Our results show that as the number of operators and events increases, LLMs tend to drift toward random guessing. For the current SOTA LLMs, we consider that their TR reasoning abilities will not exceed the boundaries imposed by the challenges that can be generated by this pipeline. Through progressively increasing the number of operators and events, we can observe the performance limits of SOTA LLMs in TR reasoning. However, since new LLMs are emerging rapidly nowadays, we acknowledge that the six models evaluated in our paper may not fully reflect the performance of the current SOTA. In this regard, we are more than willing to include additional evaluations on newer models, such as GPT-5, GPT-5 Mini, and Qwen3, in the final version.
>
> **To Comment 2** (*"The abstract reads vague and boilerplate, lacks detailed insight and lack motivation for why a new TR problem generation pipeline is needed."*)
>
> We will refine it and make the motivation clearer.
>
> **To Comment 3** (*"What is the motivation of specifically using the LTL formula for generating TR tasks, it is not clearly explained. Which families of TR problems can be represented and to what extent?"*)
>
> Thanks for pointing this out. We will explain the reason for using LTL formula to generate TR tasks in a detailed way and what sets of TR problems can be instantiated by LTL.
>
> **To Comment 4** (*"What does algorithm 1 line 4 mean? "operators ←unaryOperators+ binaryOperators" do you mean you actually add the operators, how is this defined?"*)
>
> In Line 4, “operators ←unaryOperators+ binaryOperators” means concatenating two arrays, for which we will refine and use another symbol to represent.
>
> **To Comment 5** (*"What does algorithm 1 line 5 mean? what is [[]"*)
>
> Line 5 means an initialization of an array of arrays of formulas. We will add comments to the pseudocode to make it easy to understand.
>
> **To Comment 6** (*"Section 3.4 is unclear, how are the events in Listing 3 suitable as a natural language task? Real world relations often have causal dependencies. In section 1, the authors…"*)
>
> This design is actually intentional. If the challenges were constructed based on real-world problems, they could be polluted by the common sense information and might confound the evaluation of the genuine TR ability of LLMs, making it difficult to discern whether the observed results stem from common sense knowledge or from actual TR reasoning. To mitigate this, we formulate the problems in terms of abstract events, thereby eliminating the potential interference of common sense on the assessment of TR reasoning performance. Moreover, if LLMs can demonstrate strong performance on such abstract and purely temporal reasoning tasks, in practical settings, the underlying TR reasoning ability remains intact and LLM merely replaces abstract events with real-world entities when processing TR challenges.

---

> ### Comment · Reviewer_oweF · 2025-10-20
> **Official Comment by Reviewer oweF**
>
> Thank you for your detailed response. My main concern (comment 1) with the paper remains unaddressed with the results thus hard to draw conclusions based on the provided evidences. In addition, comment 6 is not entirely convincing to me. I understand that bring sementic meanings might change LLM performance however temporal reasoning in the real world is embedded with semantic meanings thus a benchmark removing these semantics is less meaningful.  Therefore, the evidence presented in the paper as it stands remains limited.

---

> ### Author Response · Authors · 2025-10-20
>
> Thank you for your response.
>
> Regarding the **Comment 1**, we are very willing to include additional evaluations with SOTA LLMs as we mentioned in the prior response. And this process is not difficult.
>
> Regarding the **Comment 6**, we prefer to maintain our original standpoint. Actually, avoiding polluted by real-world semantic meaning when constructing synthetic datasets to assess LLMs’ reasoning ability is not rare. For example, in the work *"Test of Time: A Benchmark for Evaluating LLMs on Temporal Reasoning"* (Fatemi et al.), the authors explicitly highlight a similar standpoint like us, and in their proposed and constructed *ToT-Semantic* dataset, they also deliberately avoid introducing prior knowledge that could interfere with the evaluation and analysis of LLMs’ TR capability (you can refer to the Figure 1 and Figure 2 in their paper for reference).

---

### Decision · Action_Editor_4nsc · 2025-10-27

**Recommendation:** Reject

**Additional Comments:**

The 3 reviewers final recommendation is to not accept the paper in its current form.

Another concern raised is that the the graph generation procedure currently presented is too simplistic and should be complemented with more sophisticated processes and the natural language prompt generation is currently to superficial and its impact on model performances not enough systematically evaluated.

**Audience:**

Yes

**Audience Explanation:**

Yes, the topic is relevant to the TMLR audience.

**Claims And Evidence:**

No

**Claims Explanation:**

Despite the revisions made during the discussion period, 2 out of the 3 reviewers consider that the claims are not enough supported.

The first reviewer mentions that "the benchmark’s representativeness, the justification for the random graph generation, and the diagnostic validity of the evaluation remain insufficiently supported. The revisions are mostly descriptive and do not provide deeper analysis or additional experiments as requested.", while the second mentions that "the evidence presented in LTLBench doesn't support it as a strong useful benchmark in the community yet and further work is required", namely, adding results with more recent LLM and incorporating "additional experience to compare purely abstract reasoning vs. semantic real world reasoning where the events are actually encoded with real relations".